# CENTRALIZED TRAINING WITH HYBRID EXECUTION IN MULTI-AGENT REINFORCEMENT LEARNING

## ABSTRACT

We introduce *hybrid* execution in multi-agent reinforcement learning (MARL), a new paradigm in which agents aim to successfully perform cooperative tasks with any communication level at execution time by taking advantage of information-sharing among the agents. Under hybrid execution, the communication level can range from a setting in which no communication is allowed between agents (fully decentralized), to a setting featuring full communication (fully centralized). To formalize our setting, we define a new class of multi-agent partially observable Markov decision processes (POMDPs) that we name hybrid-POMDPs, which explicitly models a communication process between the agents. We contribute MARO, an approach that combines an autoregressive predictive model to estimate missing agents' observations, and a dropout-based RL training scheme that simulates different communication levels during the centralized training phase. We evaluate MARO on standard scenarios and extensions of previous benchmarks tailored to emphasize the negative impact of partial observability in MARL. Experimental results show that our method consistently outperforms baselines, allowing agents to act with faulty communication while successfully exploiting shared information.

## 1 INTRODUCTION

Multi-agent reinforcement learning (MARL) aims to learn utility-maximizing behavior in scenarios involving multiple agents. In recent years, deep MARL methods have been successfully applied to multi-agent tasks such as game-playing (Papoudakis et al., 2020), traffic light control (Wei et al., 2019), or energy management (Fang et al., 2020). Despite recent successes, the multi-agent setting happens to be substantially harder than its single-agent counterpart (Canese et al., 2021) because multiple concurrent learners can create non-stationarity conditions that hinder learning; the curse of dimensionality obstructs centralized approaches to MARL due to the exponential growth in state and action spaces with the number of agents; and agents seldom observe the true state of the environment.

As a way to deal with the exponential growth in the state/action space and with environmental constraints, both in perception and actuation, existing methods aim to learn decentralized policies that allow the agents to act based on local perceptions and partial information about other agents' intentions. The paradigm of *centralized training with decentralized execution* is undoubtedly at the core of recent research in the field (Oliehoek et al., 2011; Rashid et al., 2018; Foerster et al., 2016); such paradigm takes advantage of the fact that additional information, available only at training time, can be used to learn decentralized policies in a way that the need for communication is alleviated.

While in some settings partial observability and/or communication constraints require learning fully decentralized policies, the assumption that agents cannot communicate at execution time is often too restrictive for a great number of real-world application domains such as robotics, game-playing or autonomous driving (Ho et al., 2019; Yurtsever et al., 2020). In such domains, learning fully decentralized policies should be deemed inappropriate since such policies do not take into account the possibility of communication between the agents. Other MARL strategies, which take advantage of additional information shared among the agents, can surely be developed (Zhu et al., 2022).

In this work, our objective is to develop agents that are able to exploit the benefits of centralized training while, simultaneously, taking advantage of information-sharing at execution time. We introduce the paradigm of *hybrid* execution, in which agents act in scenarios with any possible communication level, ranging from no communication (fully decentralized) to full communication

between the agents (fully centralized). In particular, we focus on multi-agent cooperative tasks in which the sharing of local information (observations and actions of the agents) is *critical* to their successful execution. To formalize our setting, we start by defining *hybrid partially observable Markov decision process* (H-POMDP), a new class of multi-agent POMDPs that explicitly considers a communication process between the agents. Our goal is to find a method that allows agents to solve H-POMDPs regardless of the communication process encountered at execution time. To allow for hybrid execution, we propose an autoregressive model that explicitly predicts non-shared information from past observations of the agents. In addition, we propose a training scheme for the agents' controllers that simulates communication faults during the centralized training phase. We denote our coupled approach by *multi-agent observation sharing with communication dropout* (MARO). MARO can be easily integrated with current deep MARL methods.

We evaluate the performance of MARO across different communication levels, in different MARL benchmark environments and using multiple RL algorithms. Furthermore, we introduce three novel MARL environments that explicitly require communication during execution to successfully perform cooperative tasks, currently missing in literature. Finally, we perform an ablation study that highlights the importance of both the predictive model and the training scheme to the overall performance of MARO. The results show that our method consistently outperforms the baselines, allowing agents to exploit shared information during execution and perform tasks under various communication levels.

In summary, our contribution is three-fold: (i) we propose and formalize the setting of hybrid execution in MARL, in which agents must perform partially-observable cooperative tasks across all possible communication levels; (ii) we propose MARO, an approach that combines an autoregressive predictive model of agents' observations and a novel training scheme; and (iii) we evaluate MARO in different benchmark and novel environments, using different RL algorithms, showing that our approach consistently allows agents to act with different communication levels.

## 2 HYBRID EXECUTION IN MULTI-AGENT REINFORCEMENT LEARNING

A fully cooperative multi-agent system with Markovian dynamics can be modelled as a decentralized partially observable Markov decision process (Dec-POMDP) (Oliehoek & Amato, 2016). A Dec-POMDP is a tuple $([n], \mathcal{X}, \mathcal{A}, \mathcal{P}, r, \gamma, \mathcal{Z}, \mathcal{O})$, where $[n] = \{1, \dots, n\}$ is the set of indexes of $n$ agents, $\mathcal{X}$ is the set of states of the environment, $\mathcal{A} = \times_i \mathcal{A}_i$ is the set of joint actions, where $\mathcal{A}_i$ is the set of individual actions of agent $i$, $\mathcal{P}$ is the set of probability distributions over next states in $\mathcal{X}$, one for each state and action in $\mathcal{X} \times \mathcal{A}$, $r : \mathcal{X} \times \mathcal{A} \to \mathbb{R}$ maps states and actions to expected rewards, $\gamma \in [0, 1[$ is a discount factor, $\mathcal{Z} = \times_i \mathcal{Z}_i$ is the set of joint observations, where $\mathcal{Z}_i$ is the set of local observations of agent $i$, and $\mathcal{O}$ is the set of probability distributions over joint observations in $\mathcal{Z}$, one for each state and action in $\mathcal{X} \times \mathcal{A}$. A decentralized policy for agent $i$ is $\pi_i : \mathcal{Z}_i \to \mathcal{A}_i$ and the joint decentralized policy is $\pi : \mathcal{Z} \to \mathcal{A}$ such that $\pi(z_1, \dots, z_n) = \big(\pi_1(z_1), \dots, \pi_n(z_n)\big)$.

Fully decentralized approaches to MARL directly apply standard single-agent RL algorithms for learning each agents' policy $\pi_i$ in a decentralized manner. In independent $Q$-learning (IQL) (Tan, 1993), as well as in independent RL algorithms (Schulman et al., 2017), each agent treats other agents as part of the environment, ignoring the influence of other agents' observations and actions. More recently, under the paradigm of centralized training with decentralized execution, QMIX (Rashid et al., 2018) and other MARL algorithms (Sunehag et al., 2017; Son et al., 2019; Mahajan et al., 2019; Wang et al., 2020b; Yu et al., 2021) aim at learning decentralized policies with centralization at training time while fostering cooperation among the agents. Finally, if agents can share their observations, we can use either approach to learn fully centralized policies.

None of the classes of methods aforementioned assumes that agents may sometimes have access to other agents' observations and sometimes not. Therefore, decentralized agents are unable to take advantage of the additional information that they may receive from other agents at execution time, and centralized agents are unable to act when the sharing of information fails. In this work, we introduce hybrid execution in MARL, a setting in which agents act regardless of the communication process while taking advantage of additional information they may receive during execution. To formalize this setting, we define a new class of multi-agent POMDPs that we name hybrid-POMDPs (H-POMDPs), which explicitly considers a specific communication process among the agents.

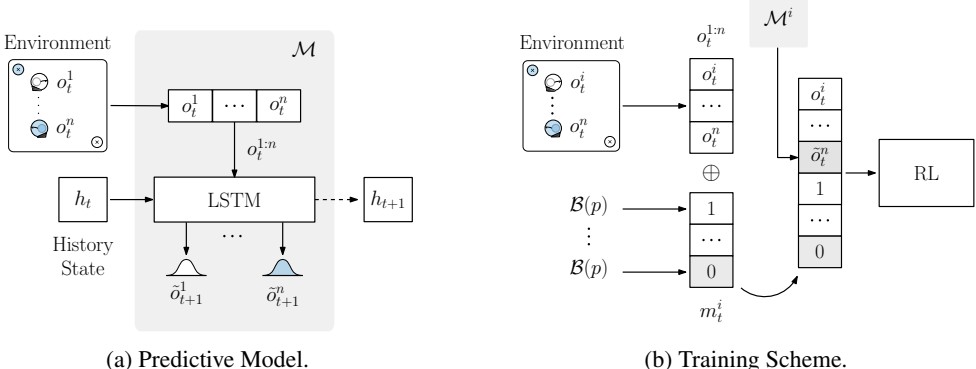

(a) Predictive Model.  (b) Training Scheme.

Figure 1: Our proposed MARO approach for hybrid execution: (a) we introduced an autoregressive predictive model $\mathcal{M}$ to estimate the next-step observations $p(o_{t+1}^{1:n} \mid o_t^{1:n}, h_t)$ from the previous ones $o_t^{1:n}$ and an history variable $h_t$; (b) we introduce a training scheme for RL controllers that randomly drops agent observations following the communication masks $m_t^i$, sampled accordingly to the communication level $p$. More details in the main text.

## 2.1 Hybrid Partially Observable Markov Decision Processes

We define an hybrid-POMDP (H-POMDP) as a tuple $([n], \mathcal{X}, \mathcal{A}, \mathcal{P}, r, \gamma, \mathcal{Z}, \mathcal{O}, C)$ where, in addition to the tuple that describes the Dec-POMDP, we consider an $n \times n$ communication matrix $C$ such that $[C]_{i,j} = p_{i,j}$ is the probability that, at a certain time step, agent $i$ has access to the local observation of agent $j$ in $\mathcal{Z}_j$. H-POMDPs generalize both the notion of decentralized execution and of centralized execution in MARL. Specifically, for a given Dec-POMDP, we can consider $C$ the identity matrix to capture fully decentralized execution and $C$ a matrix of ones to capture fully centralized execution.

In our setting, we assume that at execution time agents will face an H-POMDP with an unknown communication matrix $C$ (not provided to the agents), sampled from a set $\mathcal{C}$ according to an unknown probability distribution $\mu$. The performance of the agents is measured as $J_\mu(\pi) = \mathbb{E}_{C \sim \mu}[J(\pi; C)]$, where $J(\pi; C)$ denotes the expected discounted cumulative reward under an H-POMDP with communication matrix $C$. At training time, agents may have access to the fully centralized H-POMDP. Therefore, the setting we consider is one of centralized training with hybrid execution and unknown communication process.

We note here that every H-POMDP has a corresponding Dec-POMDP, which can be obtained by adequately changing the observation space $\mathcal{Z}$ and the set of emission probability distributions $\mathcal{O}$. Consequently, any reinforcement learning method can be trained to solve a specific H-POMDP, with a specific communication matrix $C$, by solving the corresponding Dec-POMDP. However, we seek to find a method that is able to act on H-POMDPs *regardless of the matrix $C$ that models the communication process at execution time*. To the best of our knowledge, there exists no documentation of a method that solves our problem.

## 3 Multi-Agent Observation Sharing with Communication Dropout

While acting on an H-POMDP, agents may not have access to all perceptual information due to a faulty communication process. We propose MARO, a novel approach to exploit shared information and overcome communication issues during task execution. MARO is composed of two elements: an autoregressive predictive model that estimates missing information from previous observations, and a training scheme for the RL controllers that simulates faulty communication at training time.

### 3.1 Predictive Model

The predictive model $\mathcal{M}$, depicted in Fig. 1a, is used to estimate the local observations of all agents $o_t^{1:n} = \{o_t^1, \dots, o_t^n\}$, where $o_t^i$ corresponds to the observation of the $i$-th agent, in order to overcome missing observations during execution. Thus, we learn a transition model $p(o_{t+1}^{1:n} \mid o_t^{1:n}, h_t)$ that given

the current observations $o_t^{1:n}$ and some history variable $h_t$, containing information regarding the policy of the agents, is able to predict the next-step observations $o_{t+1}$. We instantiate $p_\theta(o_{t+1}^{1:n} \mid o_t^{1:n}, h_t)$ as an LSTM, parameterized by $\theta$, with:

$$p_\theta(o_{t+1}^{1:n} \mid o_t^{1:n}, h_t) = \prod_{i=1}^{n} p_\theta(o_{t+1}^i \mid o_t^{1:n}, h_t), \tag{1}$$

where $p_\theta(o_{t+1} \mid o_t^{1:n}, h_t)$ is the Gaussian distribution of the predicted observations of the $i$-th agent. We train the predictive model and RL controllers simultaneously: we consider single-step observation transitions $(o_t^{1:n}, o_{t+1}^{1:n})$ and evaluate the negative log-likelihood of the target next-step observation $o_{t+1}^{1:n}$, given the estimated next-step observation distribution $p_\theta(\cdot \mid o_t^{1:n}, h_t)$:

$$\mathcal{L}_\mathcal{M}(o_t^{1:n}, o_{t+1}^{1:n}) = -\sum_{i=1}^{n} \log p_\theta(o_{t+1}^i \mid o_t^{1:n}, h_t). \tag{2}$$

### 3.2 Training Scheme

We also introduce an RL training scheme, depicted in Fig. 1b, which simulates the communication process at execution time and is agnostic to the type of algorithm. We setup all RL controllers to receive as input the joint observation $o_t^{1:n}$. However, in our setting, some observations may not be shared at execution time. To overcome such issue, we employ the predictive model to estimate the non-shared observations $\tilde{o}_t^j$, with $j \in [n]$. We also setup the controllers to additionally receive as input *communication masks* $m_t$, binary vectors that indicate the real and predicted components of $o_t^{1:n}$. The sequence of masks the controllers receive allows them to implicitly estimate the communication level, as well as the error that has accumulated due to long periods without receiving information from other agents, and select actions accordingly. For that, the recurrent layers are critical.

During centralized training, methods typically assume that each agent has access to the local observations of all other agents. Instead of using such information to train the agents, we instead propose to explicitly simulate the communication conditions of execution time: we randomly *dropout* agent observations. At the beginning of each episode, we sample a communication level $p \sim \mathcal{U}(0, 1)$.[1] Given $p$, we build at each time-step a communication mask $m_t^i = \{0, 1\}^n$ for each agent $i \in [n]$, with $m_t^i[i] = 1$. We sample the communication masks from independent Bernoulli distributions $m_t^i[j] \sim \mathcal{B}(p)$, for $j \in [n] \setminus i$. At execution time, we extract the observation masks $m_t^i$ directly from the environment, according to the actual faults in communication.

The communication mask indicates which components of the agent-specific joint-observation $o_t^{1:n,i}$ are dropped. Specifically, we use the real observation $o_t^k$ if $m_t^i[k] = 1$ and use the estimated observation $\tilde{o}_t^k$ otherwise. We provide each agent with an independent instance of the predictive model $\mathcal{M}^i$, which updates the estimated joint-observations in the perspective of the agent $\tilde{o}_t^{1:n,i} = \{\tilde{o}_t^{1,i}, \ldots, \tilde{o}_t^{n,i}\}$ and maintains an agent-specific history state $h_t^i$.

## 4 Evaluation

In this section, we evaluate our approach for hybrid execution, answering the following questions: **(i)** What is the performance of MARO in multi-agent cooperative tasks with partial observability, considering unknown levels of communication at execution time?; **(ii)** What is the importance of the training scheme for hybrid execution, considering different dropout schemes?; **(iii)** What is the importance of the predictive model for hybrid execution?

To address **(i)**, we evaluate in Sec. 4.3 our approach against other relevant baselines and considering multiple RL algorithms. The results show that MARO outperforms the baselines, allowing the execution of tasks across multiple communication levels. Regarding **(ii)**, we perform in Sec. 4.4.1 an ablation of the training scheme, highlighting the importance of simulating the communication process at execution time during training. We address **(iii)** in Sec. 4.4.2, highlighting the benefits of

---

[1] In the absence of prior information regarding the communication level in the environment, in the training scheme we are sampling communication matrices $C$ such that $p_{i,j} = p_{j,i} = p$ and that $p_{i,i} = 1$.

the predictive model, both in terms of training sample efficiency and in allowing centralized execution agents to exploit shared information across multiple communication levels.

## 4.1 SCENARIOS

We focus our evaluation on multi-agent cooperative environments. As discussed by Papoudakis et al. (2021b), the main challenges in current MARL benchmark scenarios, involve coordination, large action space, sparse reward and non-stationarity. As such, in these tasks the sharing of local information *is not critical* to their successful execution (as we show in Section 4.3). Thus, in order to evaluate the role of passively sharing information amongst agents in MARL, we propose three environments adapted from (Lowe et al., 2017):

- **SpreadXY**: Two heterogeneous agents cover two designated landmarks in a 2D map while avoiding collisions. In this scenario, one of the agents has access to the X-axis position and velocity of both agents, while the other agent has access to the Y-axis position and velocity of both agents. Both agents observe the landmarks' absolute position;
- **SpreadBlindfold**: Three agents cover three designated landmarks in a 2D map while avoiding collisions. Each agent's observation only includes its own position and velocity and the absolute position of all landmarks;
- **HearSee**: Two heterogeneous agents cover a single landmark in a 2D map. One of the agents ("Hear" agent) observes the absolute position of the landmark, but it does not have access to its own position in the environment. The other agent ("See" agent) observes the position and velocities of both agents, yet does not have access to the position of the landmark.

In addition to the proposed environments, we evaluate our approach in standard MARL benchmark scenarios, in particular in the SpeakerListener environment. For a complete description of the scenarios, we refer to Appendix B.1.

Finally, we consider H-POMDPs with communication matrices such that each agent $i$ can always access its own local observation, i.e., $p_{i,i} = 1$, and the communication matrix is symmetric between agents $i$ and $j$, i.e., $p_{i,j} = p_{j,i}$. To simplify the exposition and the evaluation, we use the same $p_{i,j} = p$ for all pairs of different agents $i$, $j$. Therefore, we also use $p$ to unambiguously denote the communication level of a given H-POMDP. We perform a comparative study between different sampling schemes for the communication matrix in Appendix B.3.3, highlighting the robustness of MARO under different communication settings. Additionally, we note that in all the environments the previous action taken by agent $i$, $a_{t-1}^i$, is included in its local observation $o_t^i$.

## 4.2 BASELINES AND EXPERIMENTAL METHODOLOGY

We compare MARO against different baselines that correspond to different levels of information-sharing between the agents. We consider two "extreme" cases:

- **Observation** (Obs.): Agents only have access to their own observations and are unable to communicate with other agents, corresponding to standard MARL algorithms designed for decentralized execution;
- **Joint-Observation** (J. obs.): Agents always have access to the observations of all agents, corresponding to standard MARL algorithms designed for centralized execution. This baseline is unable to perform when communication fails and can be seen as an upper bound.

To the best of our knowledge, there exists no method developed specifically for the problem of executing with faulty communication with unknown dynamics to serve as a direct comparison to MARO. As such, we adapt the model proposed by Kim et al. (2019b) as a baseline:

- **Message-Dropout** (MD): Agents train with communication failing half of the times (fixed $p = 0.5$), without communication masks and without the predictive model.

We employ the same controller networks across all evaluations. The networks include recurrent layers to mitigate the effects of partial observability. We consider two different MARL algorithms: QMIX and Independent $Q$-Learning (IQL). We follow the training hyperparameters suggested by

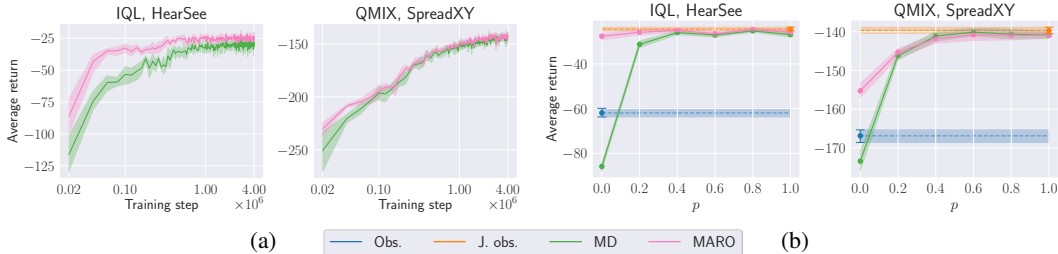

(a)          Obs.    — J. obs.    — MD    — MARO          (b)

Figure 2: (a) Average episodic returns during training with 95% confidence interval for MARO and MD; (b) Average episodic returns during training with 95% confidence interval for different communication levels at execution time for all approaches. Best viewed with zoom.

Table 1: Average episodic returns and 95% bootstrapped confidence interval over five seeds for all approaches in all scenarios. Higher is better.

| Environment | IQL | | | | QMIX | | | |
|---|---|---|---|---|---|---|---|---|
| | Obs. | J. obs. | MD | MARO | Obs. | J. obs. | MD | MARO |
| SpreadXY | -173.2 (-0.8,+1.1) | -140.3 (-0.8,+0.8) | **-150.6** (-1.1,+1.6) | **-148.7** (-0.7,+0.7) | -166.9 (-1.7,+1.5) | -139.7 (-0.8,+0.9) | -147.1 (-0.5,+0.5) | **-144.2** (-0.5,+0.6) |
| SpreadBlindfold | -432.2 (-5.9,+6.0) | -403.4 (-5.4,+3.9) | **-402.7** (-1.0,+1.0) | **-405.3** (-4.3,3.4) | -418.1 (-3.3,+3.5) | -376.0 (-2.9,+3.3) | **-405.7** (-5.0,+5.0) | **-401.2** (-6.4,+6.4) |
| HearSee | -61.9 (-1.8,+2.1) | -24.5 (-0.8,+0.9) | -37.1 (-0.6,+0.6) | **-25.9** (-0.3,+0.3) | -54.9 (-1.5,+1.3) | -24.1 (-0.7,+0.9) | **-25.4** (-0.4,+0.4) | **-25.1** (-0.3,+0.4) |
| SpeakerListener | -31.7 (-1.2,+1.4) | -25.4 (-1.1,+1.1) | -27.8 (-0.3,+0.3) | **-25.6** (-0.5,+0.6) | -26.1 (-1.3,+1.2) | -25.1 (-1.1,+1.1) | **-23.2** (-1.0,+1.2) | **-23.3** (-1.1,+1.3) |

Papoudakis et al. (2021b); we train all models for 4M steps, performing 5 training runs for each experimental setting and 50 evaluation rollouts for each training run. We assume that $p = 1$ at $t = 0$ for the MD and MARO algorithms. The performance of the Obs. and J. Obs. agents is evaluated by aggregating evaluation rollouts with $p = 0$ and $p = 1$, respectively. The other algorithms are evaluated for $p$ sampled from a discretized uniform distribution. If the communication level is not explicitly referred, then the values correspond to the average performance across all communication levels. We refer to Appendix B for a complete description of the experimental methodology, including hyperparameters of the predictive model and the RL controllers. Our code is available on Github.

## 4.3   RESULTS

We present the main evaluation results in Table 1. For each environment, RL algorithm and method, we present the values of the accumulated rewards obtained. The values that are not significantly different than the highest are presented in bold. The results show that MARO consistently performs equal or better than the MD baseline across all scenarios and algorithms. MARO is able to exploit the information provided by the other agents, in contrast with the fully decentralized approaches (Obs.). Moreover, MARO is often able to achieve performances comparable to the fully centralized agent (J. Obs.), which executes with full communication, despite failures in communication. We also note here that, in some MARL environments, information sharing between agents may not lead to performance gains. For instance, in the SpeakerListener scenario, decentralized QMIX (Obs.) is able to perform competitively in comparison to centralized QMIX (J. Obs.), without requiring information sharing.

In Fig. 2, we highlight the training curves and the performance of the approaches for different communication levels $p$ for the HearSee and SpreadXY environments (more in Appendix B). The results show that MARO outperforms the MD baseline in terms of sample efficiency, with a bigger jump-start in the initial stages of the training (Fig. 2a), and overall performance across all communication levels (Fig. 2b). Additionally, MARO significantly outperforms the Obs. baseline in settings with no communication ($p = 0$); MD struggles to act in the same setting, performing worse than the fully decentralized baseline. Moreover, the performance of MARO improves as the level of communication in the environment increases, showing that our model is able to efficiently make use of all provided information. Appendix B includes all training curves and performance results.

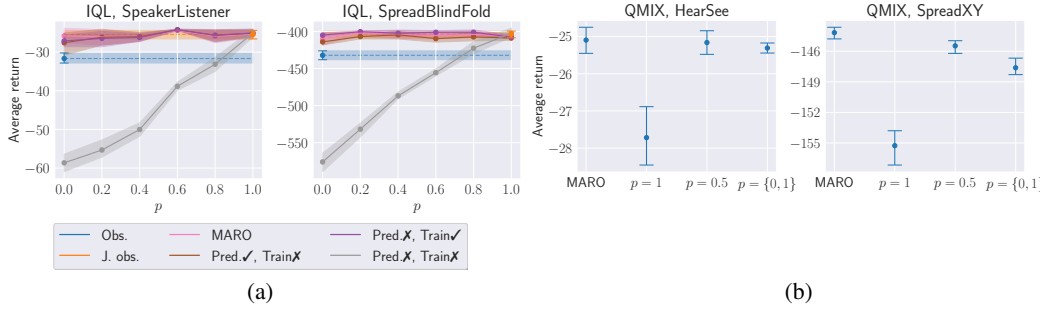

(a)                                                                                    (b)

Figure 3: (a) Average episodic returns with 95% confidence interval for different communication levels at execution time for the ablated versions of MARO; (b) Average episodic returns with 95% confidence interval for different sampling schemes. Best viewed with zoom.

Table 2: Average episodic returns and 95% bootstrapped confidence interval over five seeds for all versions of MARO in all scenarios. Higher is better.

| | IQL | | | | QMIX | | | |
|---|---|---|---|---|---|---|---|---|
| **Environment** | **MARO** | **Pred. ✓ Train ✗** | **Pred. ✗ Train ✓** | **Pred. ✗ Train ✗** | **MARO** | **Pred. ✓ Train ✗** | **Pred. ✗ Train ✓** | **Pred. ✗ Train ✗** |
| SpreadXY | -148.7 (-0.7,+0.7) | -158.1 (-0.6,+0.6) | **-146.4** (-0.6,+0.7) | -197.4 (-1.7,+1.8) | **-144.2** (-0.5,+0.6) | -155.3 (-1.5,+1.9) | **-144.4** (-1.1,+1.7) | -197.8 (-1.7,+1.7) |
| SpreadBlindfold | **-405.3** (-4.3,3.4) | -408.6 (-3.0,+3.0) | **-402.9** (-1.8,+2.1) | -479.2 (-2.0,+2.4) | **-401.2** (-6.4,+6.4) | -393.8 (-5.2,+4.8) | -407.7 (-5.9,+6.9) | -479.6 (-3.9,+4.3) |
| HearSee | **-25.9** (-0.3,+0.3) | -27.7 (-0.5,+0.5) | **-26.1** (-0.3,+0.3) | -78.0 (-4.2,2.6) | **-25.1** (-0.3,+0.4) | -27.7 (-0.8,+0.7) | **-25.1** (-0.3,+0.3) | -79.4 (-3.4,+3.4) |
| SpeakerListener | **-25.6** (-0.5,+0.6) | **-25.8** (-0.7,+0.7) | **-25.8** (-0.7,+0.7) | -43.5 (-0.9,+0.9) | **-23.3** (-1.1,+1.3) | **-23.5** (-1.2,+1.4) | **-23.3** (-1.2,+1.4) | -39.0 (-1.5,+1.7) |

## 4.4 ABLATION STUDY

We perform an ablation study on the two components of MARO to study their impact on the overall performance of the method. We introduce ablated versions of MARO along two different axes: (i) training with our proposed scheme (**Train ✓**), against training with fixed $p = 1$ (**Train ✗**); and (ii) using the predictive model to estimate missing observations (**Pred ✓**), against a naive dropout approach that replaces missing observations with zeros (**Pred ✗**). Table 2 shows the results of the ablation study across environments and algorithms. We discuss the results along each axis separately.

### 4.4.1 TRAINING SCHEME

The results in Table 2 highlight the importance of the training scheme for hybrid execution: introducing our proposed training scheme (**Pred ✗, Train ✓**) over the fully-ablated version of MARO (**Pred ✗, Train ✗**) results in a significant performance improvement, while removing the training scheme from MARO (**Pred ✓, Train ✗**) reduces the performance of our approach.

In Fig. 3a we evaluate the impact of the training scheme in the performance across different communication levels, shown for the SpeakerListener and SpreadBlindfold environments (additional results in Appendix B). The results reveal that MARO is able to perform well across the whole communication spectrum. The results also show that while the fully-ablated version of our approach struggles to act with any communication level other than $p = 1$, adding MARO's training scheme alone results in a method that is able to perform close to optimally across the entire spectrum of $p$. Finally, removing the training scheme from MARO results in a performance drop, especially for $p = 0$. In summary, the training scheme is beneficial for different values of $p$ without sacrificing performance for $p = 1$.

We also evaluate the impact of different sampling strategies of $p$ during training. As previously explained, our proposed communication sampling scheme has $p \sim \mathcal{U}(0, 1)$ during training. In Fig. 3b, we compare our sampling scheme in the HearSee and SpreadXY environments (more in Appendix B) against three other sampling approaches: a categorical distribution over the communication level

extremes, $p \sim \mathcal{U}\{0, 1\}$; a fixed sampling scheme with $p = 0.5$, which corresponds to the training approach of Kim et al. (2019b) with additional observation masks; and a fixed sampling scheme with $p = 1$, without observation masks. The results highlight the importance of simulating faulty communication during the training phase, as the fully centralized training scheme ($p = 1$) is outperformed by all other approaches. In Appendix B, we present the full results of this evaluation, showing the impact of the sampling scheme in other scenarios and RL algorithms.

### 4.4.2 PREDICTIVE MODEL

The results in Table 2 also highlight the significant improvement in the performance of MARO when employing the predictive model to estimate the missing observations. Specifically, adding the predictive model (**Pred ✓, Train ✗**) to the fully ablated version of MARO (**Pred ✗, Train ✗**) results in a significant performance increase. Removing the predictive component from MARO (**Pred ✗, Train ✓**) still results in a competitive algorithm without a significant performance drop in the majority of scenarios. However, there are significant advantages to employing the predictive model:

- The predictive model improves the sample efficiency of MARO. In Fig. 4c, we show the training curves for MARO, with and without the predictive model. The results show that employing the predictive model results in a significant jump-start in terms of sample efficiency. The predictive model provides the RL controllers with an estimate of the missing agent trajectories, instead of replacing the missing input with zeros, resulting in improved performance during the initial stages of the training. This improvement is consistent across several environments and algorithms, as shown in Appendix B.

- The predictive model provides robustness to centralized execution methods when performing tasks in settings with potential faulty communication, despite never being trained to execute in such conditions. In other words, the predictive model allows for zero-shot multi-agent execution with respect to the communication failures. In Fig. 4d, we show that employing the predictive model to predict missing information at execution time allows a standard centralized method to perform the task with minimum performance loss (**Pred ✗, Train ✗**).

- The predictive model is able to perform accurate agent modelling with faulty communication, providing an interpretable insight into the decision-making process of the agents. In Fig. 4a, we show the predicted trajectories of all agents in the perspective of the blue agent, which are close to the real trajectories performed by all agents (more in Appendix B.3.5).

We can also assess the correctness of the predictions made by the predictive model by evaluating the performance of MARO against a *Switch* baseline. In this baseline, the agents choose actions using two controllers, selected accordingly to the level of communication in the environment at each timestep: one that uses the joint observation at the timesteps it is available (similar to the Joint Observation baseline), and one that uses only the local observation, otherwise (similar to the Observation baseline). We show the results of the comparison in performance between MARO and the Switch baseline in Fig. 4b. The results reveal that MARO is able to exploit the predicted observations for communication levels $p < 1$, outperforming the Switch baseline in different communication settings.

## 5 RELATED WORK

In this section, we connect our work with other lines of research, discussing the similarities and differences between our study and previous works in the field. Due to space limitations, we discuss only the most relevant works and provide an extended discussion of related work in Appendix A.

Closely related to our work are studies that address the problem of partial observability in MARL. As an example, Omidshafiei et al. (2017) propose a decentralized MARL algorithm that uses RNNs to improve the agents' observability. Mao et al. (2020) use an RNN to compress the agents' histories, helping to improve agents' observability. The commonly used paradigm of centralized training with decentralized execution also contributes to alleviating partial observability at training time (Oliehoek et al., 2011; Rashid et al., 2018; Foerster et al., 2016; 2017; Sunehag et al., 2017; Son et al., 2019; Mahajan et al., 2019; Wang et al., 2020b; Yu et al., 2021).

Other lines of research investigate communication techniques for MARL (Zhu et al., 2022), focusing on how (Niu et al., 2021; Kim et al., 2019a), when (Singh et al., 2018; Hu et al., 2020), and what

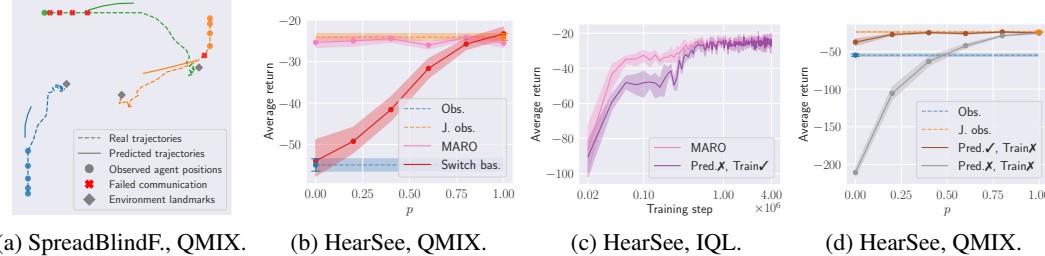

(a) SpreadBlindF., QMIX.  (b) HearSee, QMIX.  (c) HearSee, IQL.  (d) HearSee, QMIX.

Figure 4: Evaluation of the predictive model of MARO: (a) Trajectory estimation from the perspective of the blue agent, using its agent-specific predictive model; (b) Average episodic returns with 95% confidence interval for different communication levels at execution time of MARO against the Switch baseline; (c) Average episodic returns during training with 95% bootstrapped confidence interval for MARO and an ablated version without the predictive model; (d) Average episodic returns with 95% confidence interval for different communication levels at execution time of ablated versions of MARO, showing the ability of the predictive model for zero-shot execution. Best viewed with zoom.

(Foerster et al., 2016) to communicate to foster cooperation. Previous studies focused on the sharing of (encoded) local observations and actions among agents (Foerster et al., 2016) in a proxy-like manner (Wang et al., 2019). Others learn mechanisms that improve communication efficiency by either limiting the variance of exchanged messages (Zhang et al., 2019) or temporally smoothing information shared between agents (Zhang et al., 2020). Kim et al. (2019b) propose message-dropout, which aims at making learning robust against communication errors. Message-dropout drops the messages received from other agents independently at random during training before inputting them into the RL algorithm. In a similar fashion to message dropout, Wang et al. (2020a) propose a recurrent actor-critic algorithm for handling multi-agent coordination with limited communication, showing that recurrence successfully contributes to robust performance under communication failures.

In contrast, we assume that agents have no control over when and with whom to communicate. Hence, they should robustly perform under any type of communication policy/level at execution time. Also, we do not learn the content of the messages and consider a rather passive communication setting in which agents share local observations and actions. For this reason, we did not include the works of Zhang et al. (2019) and Zhang et al. (2020) as baselines since the comparison between methods would not be meaningful. Instead, following both Kim et al. (2019b) and Wang et al. (2020a), we use message-dropout with recurrent learners as a baseline. Finally, none of the aforementioned works proposed the use of predictive models to account for missing information at execution time as we do in our work, nor mathematically formalized hybrid execution as we present in Sec. 2.1.

## 6 CONCLUSION

In this work, we introduce *hybrid* execution, a new paradigm in which agents act under any communication level at execution time, while exploiting information-sharing among the agents. To formalize our setting, we define hybrid-POMDPs, a new class of POMDPs that explicitly considers a communication process between the agents. To allow for hybrid execution we propose MARO, a novel approach that combines an autoregressive predictive model, to estimate missing observations, and an RL training scheme that randomly drops agent observations, simulating different communication levels during the centralized training phase. We show that MARO allows agents to act across different communication levels, successfully exploiting available shared information.

Future work could comprise: (i) evaluating MARO with actor-critic RL algorithms; (ii) studying other neural network architectures, such as graph neural networks, for better multi-agent trajectory prediction; (iii) studying the performance of MARO under other communication settings.

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
