# OpenReview forum: "Centralized Training with Hybrid Execution in Multi-Agent Reinforcement Learning"
_ICLR.cc/2023/Conference — Submitted to ICLR 2023_

### Official Review · Reviewer_CnPR · 2022-10-24

**Confidence:** 5
**Clarity, Quality, Novelty And Reproducibility:** 1. The clarity of presentation is goo…
**Correctness:** 4
**Technical Novelty And Significance:** 2
**Empirical Novelty And Significance:** 2
**Recommendation:** 5

**Strength And Weaknesses:**

Strength
1. The paper is well written in terms of describing the setting used.
2. The hybrid execution setting is moderately relevant for real applications.

Weakness
1. The related works is incomplete. The paper lacks discussion about the implications of using the different types of MARL algorithms: decentralized/centralized and policy/value based and their interaction with the hybrid communication setting, which is important (refer to e.g. [1], [2]). Policy based MARL methods like Tesseract [2] and MAPPO [3] can handle this scenario without performance loss in principle. Discussion about the mentioned works should be included for better context and reader understanding.

2. The experimental results are reported only on toy domains, there are more realistic complex domains like Starcraft which can be used. The baselines used are also inadequate, some policy based baselines should be included (for reasons covered in [1, 2]). Even for value based setting, there are stronger baselines available like [4].

3. The difference with performance of the simple message dropout baseline is not significant, which makes it unclear how effective the proposed approach is.

4. Is the communication matrix C also visible to the agents?

5. In general, the considered setting is significantly easier than decentralized control, so I am unconvinced with the usefulness of the approach.

References:
1. Maven: Multi-agent variational exploration, Mahajan et al, 2019
2. Tesseract: Tensorised Actors for Multi-Agent Reinforcement Learning, Mahajan et al 2021
3. The Surprising Effectiveness of PPO in Cooperative Multi-Agent Games, Yu et al 2021
4. Rode: Learning roles to decompose multi-agent tasks, Wang et al, 2021

**Summary Of The Paper:**

This paper proposes a method for training in MARL setting under hybrid communication setting (decentralized vs fully centralized) by using a autoregressive model for predicting next joint observations. Experiment are provided on some toy domains with 2 decentralized value based baselines.

**Summary Of The Review:**

Interesting idea, but the results are not strong. The paper also lacks important related works crucial for better understanding the problem. The authors should focus on the points in the weakness section.

---

> ### Author Response · Authors · 2022-11-14
> **Response to reviewer CnPR**
>
> ## Additional related work
>
> We thank the reviewer for indicating works that could improve the introduction of the paradigm of hybrid execution and the discussion of related work. We include the corresponding changes in the updated version of our manuscript. We note that, due to lack of space in the main text, we moved related work regarding multi-agent trajectory prediction to Appendix A.
>
> We also clarify here that our contributions are agnostic to specific MARL methods for decentralized execution. In this work, we chose to the use the value-based MARL methods IQL and QMIX with MARO but other MARL methods could be used, as the reviewer points out. We argue that our focus is on making MARL algorithms, in general, apt for hybrid execution, rather than MARL algorithms themselves.
>
> ## Realistic complex environments (SMAC)
>
> We thank the reviewer for the comment. The problem of hybrid execution itself is a novel contribution of our work. Consequently, some (or most) of the existing benchmark environments for MARL are not perfectly suited for our evaluation. Specifically, on the Starcraft environment, we saw no performance gain from information sharing between agents at execution time, since the decentralized algorithms already achieve the maximum possible win rate. In fact, we think of this fact as a limitation of current MARL benchmark environments: as the problem of decentralized execution became central to research, the benchmark environments benefit more from coordinate behavior than information sharing. In other words, and specifically, a good centralized execution method performs equally well as a good decentralized execution method ($100\%$ win rate) and, consequently, there is no advantage in using hybrid execution methods instead of decentralized execution methods. We included the SMAC results in the table below.
>
> |           |        Obs.        |        J.Obs       |
> |---------|------------------|------------------|
> |   Return  | 19.7 (-0.02,+0.01) | 19.6 (-0.00,+0.01) |
> | Win rate  | 0.95 (-0.03,+0.02) | 0.93 (-0.01,+0.01) |
>
> Table 1: Average return and win rate (alongside 95 % bootstrapped confidence interval) over five seeds for Obs and J.Obs in the SMAC (2s3z) environment (IQL algorithm). Higher is better.
>
> ## Evaluation with different RL methods
>
> We thank the reviewer for suggesting that using more MARL methods for decentralized execution such as MAPPO, Tesseract and RODE would strengthen our work. Even though we agree with the reviewer in that the referred methods could technically be included in our experimental evaluation, we argue that such inclusion would only add so much regarding the focus of our work. The focus of our work is on the perception of the multiple agents, as well as the training scheme of RL methods to prepare them for hybrid execution. Our focus is complementary to any effort on decentralized MARL algorithms themselves such as the ones the reviewer suggests. Additionally, our MARO method can be used in combination with any algorithm for decentralized execution, including the ones the reviewer suggests.
>
> ## Comparison with MD baseline
>
> As the reviewer rightfully identifies, for some of the environments and MARL algorithms used, our proposed MARO method does not significantly improve over the chosen baseline MD. At the same time, we highlight that (i) our MARO method is **never** significantly worse than the MD baseline and that it is significantly better for **some** of the environments and MARL algorithms. Then, (ii) even though we use it as a baseline,  MD was not developed and had never been used for the purpose of hybrid execution that we introduce in this work. Finally, (iii) MARO brings other empirical advantages over MD, the most evident of those being improved sample efficiency (Figure 2(a)). We elaborate on the advantages in Section 4.4.2.
>
> Additionally, our latest results on SpreadXY4 indicate that our proposed MARO scales better than MD in an environment with more agents. We refer to the second table in the response to reviewer PDRN for the results.
>
> ## Notes on $C$
>
> We thank the reviewer for the question. We note that the communication matrix $C$ is not known by the agents. We clarify this in Section 2.1 of the updated version of the paper. Additionally, the matrix $C$ is not the same across episodes. In other words, in hybrid execution agents must be able to act regardless of the matrix $C$ of the environment at execution time.

---

> > ### Author Response · Authors · 2022-11-14
> > **Response to reviewer CnPR (continues)**
> >
> > ## On the difficulty of hybrid execution
> >
> > We respectfully disagree with the reviewer in that the hybrid execution setting we introduce is easier than the decentralized execution setting. We call attention for the fact that hybrid execution is the problem where at execution time agents may face full decentralization, full centralization, or any other degree of information-sharing between these two extreme cases. As such, the problem of hybrid execution strictly includes the problem of decentralized execution. Consequently, the problem of hybrid execution is strictly harder than the problem of decentralized execution.
> >
> > ---
> >
> > **We hope that our response has addressed the reviewer's concerns. Please let us know if additional clarifications are required to improve the score.**

---

### Official Review · Reviewer_eHqC · 2022-10-24

**Confidence:** 4
**Correctness:** 3
**Technical Novelty And Significance:** 3
**Empirical Novelty And Significance:** 3
**Recommendation:** 5

**Clarity, Quality, Novelty And Reproducibility:**

The paper is clearly written and well-presented. The paper contributes both a novel framework (H-POMDP), as well as a new training scheme for MARL (MARO) and the code is open-sourced.


**Strength And Weaknesses:**

Strengths:
- The work formalises the setting of faulty communication of the agent's observations during execution as a hybrid-POMDP (H-POMDP)
- The work introduces a novel training approach, MARO, that can be combined with many existing MARL methods to improve performance in this faulty communication setting


Weaknesses:
- Lack of motivation for selected benchmarks. In the abstract it is mentioned "tailored to emphasize the negative impact of partial observability", but I did not find an explanation for supporting this statement thereafter.
- Weak comparison against the selected method, can be improved by clarifying more details on the MD approach

Q1. Can you motivate the selection of the benchmark, in contrast for example with the choice of the Kim et al. (2019b) work. Why are they "tailored to emphasize the negative impact of partial observability"?

Q2. As far as I can tell, there are never more than three agents in the environment. Have you also tried larger environments? Can you comment on the scalability of the method? Especially in combination with independent learning approaches, I do not see why a larger number of agents might pose an issue. But what about the predictive model?

Q3. Have you tried combining MARO with PG/actor-critic based approaches? It would have been interesting and make for a well-rounded experimental evaluation to also see for example how IPPO behaves.

Q4. Can you clarify if for the adapted MD approach you also used the original training procedures (eg., block-wise/element-wise dropout)? Have you tried also other values for p?

Q5. In 3.2 you mention that the masks enables to measure uncertainty regarding the input. Is this information used anywhere?

Q6. Have you also tested a setting where during execution the p is dynamic?


Minor typo:
- Section 6 'that explicity considers'

**Summary Of The Paper:**

In the context of MARL, this work introduces the paradigm of centralised training with hybrid execution, a setting in which the information availability of the other agent's observation is not guaranteed during execution. A novel training approach is also introduced, MARO, in order to deal with this setting, comprising of a predictive model and a training scheme that includes a random drop of agent observations, to simulate the conditions that might be encountered during execution.

**Summary Of The Review:**

All in all this, this work addresses a novel setting, that of faulty observations exchange between agents during execution, and proposes a novel framework to capture this setting and a novel training approach for solving it. I would consider it currently slightly below the acceptance threshold, but if authors can clarify my questions, I will raise my score.

---

> ### Author Response · Authors · 2022-11-14
> **Response to reviewer eHqC**
>
> We thank the reviewer for the comments and for acknowledging the novelty of our problem and our approach.
>
> ---
>
> ## Q1 - Motivation for our environments
> In this work, we explore multi-agent cooperative tasks in which the sharing of information is \emph{critical} to their successful execution. To that end, we contributed with three scenarios in which information sharing is fundamental for task execution due to the intrinsic partial observability of the environment:
> - In **SpreadXY**, each agent only observes the position and velocity (of both agents) along one dimension; only through shared information can the agents localize themselves in the 2D environment to efficiently cover the landmarks, avoiding collisions.
> - In **SpreadBlindfold**, each agent only observes its own position and velocity; only through shared information can the agents avoid collisions and efficiently cover all the landmarks. This scenario is similar to the "Cooperative navigation" scenario of Kim et al. (2019b).
> - In **HearSee**, one agent observes the position and velocity of both agents while the other agent observes the position of the landmark; only through shared information can both agents reach the landmark.
>
> These scenarios can be considered extensions of previous MARL benchmark scenarios, recently evaluated by Papoudakis et al (2019b), which are easily interpretable and efficient to train. We clarified the motivation behind the creation of the new benchmark environments in Section 4.1 of the updated manuscript.
>
> ## Q2 - More agents
>
> Following the reviewer's suggestion, we have evaluated MARO in a scenario with a higher number of agents: we considered an extension of SpreadXY with 4 agents (SpreadXY4).
>
> |          Obs         |        J. Obs        |          MD         |         MARO        |
> |:--------------------:|:--------------------:|:-------------------:|:-------------------:|
> | -1194.5 (-9.0,+12.0) | -1099.4 (-11.3,+4.1) | -1176.9 (-5.7,+7.1) | -1152.6 (-4.9,+5.0) |
>
> Table 1: Average episodic returns and 95\% bootstrapped confidence interval over five seeds for all approaches in the SpreadXY4 environment for IQL. Higher is better.
>
> The results in the above table show that MARO is able to outperform the MD baseline and exploit passively shared information to perform the task, even in scenarios with higher number of agents.
>
> Additionally, we also evaluated the suitability of SMAC (with 5 agents) as a scenario to evaluate approaches for hybrid execution, i.e., as a scenario where the passive sharing of information amongst the agents brings any benefit to task execution.
>
> |           |        Obs.        |        J.Obs       |
> |---------|------------------|------------------|
> |   Return  | 19.7 (-0.02,+0.01) | 19.6 (-0.00,+0.01) |
> | Win rate  | 0.95 (-0.03,+0.02) | 0.93 (-0.01,+0.01) |
>
> Table 2: Average return and win rate (alongside 95\% bootstrapped confidence interval) over five seeds for Obs and J.Obs in the SMAC (2s3z) environment (IQL algorithm). Higher is better.
>
> However, the results in the above table show that there is no significant performance gap between the fully centralized (Jobs) and fully decentralized (Obs) agents. We already discussed the same phenomenon in the original manuscript, in regards to the result of the agents in the Speaker-Listener environment with QMIX algorithm (Table 1, Page 6). This absence of a performance gap highlights the fact that, in some tasks, there is no benefit to having access to shared information for the decision-making of the agents. In contrast, in this work we focus on multi-agent cooperative tasks in which the sharing of information is \emph{critical} to their successful execution.
>
> ## Q3 - Different RL methods (IPPO)
>
> We thank the reviewer for the comment. In this work, we propose MARO, an approach to allow for hybrid execution. We agree with the reviewer that IPPO (and many other RL algorithms) could also be employed to evaluate MARO. However, our claim that MARO allows agents to exploit shared information for hybrid execution is, in itself, agnostic to the downstream RL algorithm employed. We already explored such agnosticism by evaluating both with IQL and QMIX algorithms: a fully decentralized method and a centralized training with decentralized execution method. For both methods, we highlighted the performance of MARO against the baseline approach. As IPPO falls into the first category, we believe that additional evaluation would improve the evaluation section, yet would not validate any claim that was not already been validated with the current selection of MARL algorithms.
>
> ## Q4 - Details regarding the MD baseline
>
> We employ the same dropout training scheme as in MARO (block-wise). We follow the original formulation of Kim et al, (2019b), and use $p=0.5$ throughout the evaluation.

---

> > ### Author Response · Authors · 2022-11-14
> > **Response to reviewer eHqC (continues)**
> >
> > ## Q5 - Clarify how masks measure uncertainty and how that is useful for RL
> >
> > The information provided by the masks is used for the RL policies to implicitly estimate the degree of observability of the environment, i.e., $p$, and act accordingly. For example, if no communication is received for a long time, the RL controller can estimate that most of the other agents' information are predictions from the predictive model, not actual observations, and are therefore most likely noisy. Consequently, the RL agent may choose to use a more decentralized-like policy. The reverse happens if for a long time the masks indicate that real observations of the other agents are being communicated. Empirically, the necessity of the masks can be asserted by looking at the difference in performance between the MD baseline in Table 1 and the ablated version of MARO without predictive model in Table 3. Finally, we clarify that no communication masks are provided to the prediction model, only to the reinforcement learning policy.
> >
> > ## Q6 - Evaluation with dynamic communication process
> >
> > We thank the reviewer for suggesting that including tests with dynamic (non-stationary) communication processes in hybrid execution would strengthen our work. Following the reviewer's suggestion, as well as the comments of reviewer PDRN, we performed additional experiments regarding dynamic communication processes.  We considered three different settings: a) our default setting $p_{\textrm{default}}$, where $p_{i,j} = p_{j,i} = p$, with $p \sim U(0,1)$ sampled at the beginning of each episode; b) $p_{\textrm{asymmetric}}$, with communication matrices $C$ such that $p_{i, j} \neq p_{j, i}$, where both $p_{i, j}$ and $p_{j, i}$ are independently sampled from a random uniform distribution at the beginning of each episode; c) $p_{\textrm{dynamic}}$, with dynamic communication matrices $C$ that are sampled every $5$ time steps. We present the results of this comparison in Appendix B.3.3, due to lack of space on the main text. We reproduce here a representative selection of the results, for convenience.
> >
> > |                           |         SpreadXY + QMIX        |       SpreadBlindf. + IQL      |
> > |:-------------------------:|:------------------------------:|:------------------------------:|
> > |   $p_{\textrm{default}}$  | -143.0 (-144.9, -141.4) | -405.0 (-407.4, -403.4) |
> > | $p_{\textrm{asymmetric}}$ | -141.6 (-143.1, -140.3) | -398.6 (-412.2, -383.3) |
> > |   $p_{\textrm{dynamic}}$  | -141.0 (-143.1, -139.0) | -403.6 (-413.6, -395.5) |
> >
> > Table 3: Average episodic returns and 95% bootstrapped confidence interval over five seeds for different sampling techniques for communication matrices in MARO. Higher is better.
> >
> > The results highlight the robustness of MARO to different sampling choices of the communication matrices between the agents, with no significant performance difference between the three conditions.
> >
> > ---
> >
> > **We hope that our response has addressed the reviewer's concerns. Please let us know if additional clarifications are required to improve the score.**

---

### Official Review · Reviewer_ymGd · 2022-10-26

**Confidence:** 4
**Correctness:** 3
**Technical Novelty And Significance:** 2
**Empirical Novelty And Significance:** 2
**Recommendation:** 5

**Clarity, Quality, Novelty And Reproducibility:**

-the paper is well written
-the novelty may be limited as it builds on Dec-POMDPs
-more complex agent behavior, especially dynamic agent interactions needs to be modeled
-the notion on how uncertainty is captured in the centralized training scheme needs to be explained


**Strength And Weaknesses:**

strengths:
-the paper proposes an approach called hybrid-POMDPs
-it adds the notion of agent communications to dec-POMDPs which is fully decentralized
-agents can have any level of communication (none to complete)


weakness:
-the contribution may be limited because when the communication matrix is known it can be solved as a Dec-POMDP
-predicting the unknown observations using an LSTM does not consider all possible contextual information to identify dynamic agent behavior
-the centralized training scheme needs to be explained in more detail
-how accurately masking during training reflects agent communication during evaluation needs to be discussed
-the MD approach used for comparison performs equally better in many scenarios


**Summary Of The Paper:**

The paper proposes multi-agent hybrid-POMDPS which uses a centralized training scheme as well as models a communication process between the agents. Experiments are conducted on standard benchmarks to show the superiority of the proposed method.

**Summary Of The Review:**

The paper introduces hybrid-POMDPs by adding agent communication as an additional parameter to dec-POMDPs. However, the novelty and contribution is limited as the autoregressive prediction model and the masked training procedure does not consider exhaustive multi-agent scenarios. The experiments also show that another baseline performs similar using IQL. The main advantage of the proposed approach over existing works needs to be made clear via experimentation.

---

> ### Author Response · Authors · 2022-11-14
> **Response to reviewer ymGd**
>
> We start by thanking the reviewer for the comments and for acknowledging that we contribute the novel paradigm of hybrid execution in MARL, by extending the traditional Dec-POMDP model with an explicit communication process between agents. We highlight here that, as an added difficulty in our work, at execution time the agents may face a Hybrid-POMDP with **any** communication process. We address the reviewer's concerns below.
>
> ---
>
> ## Difference between Hybrid-POMDPs and Dec-POMDPs
>
> As the reviewer correctly points out, for each communication process $C$, an Hybrid-POMDP has a corresponding Dec-POMDP that can be solved by traditional MARL methods. However, it is important to clarify here that the problem we tackle in our work is not the one of solving a specific hybrid-POMDP with a specific communication process $C$. In our work we tackle a harder problem: learning multi-agent policies that can be used for any communication process. We elaborate this discussion in the third paragraph of Section 2.1 in the updated manuscript.
>
> ## Comparison with MD baseline
>
> As the reviewer rightfully identifies, for some of the environments and MARL algorithms used, our proposed MARO method does not significantly improve over the chosen baseline MD. At the same time, we highlight that (i) our MARO method is **never** significantly worse than the MD baseline and is significantly better for **some** of the environments and MARL algorithms; (ii) even though we use it as a baseline,  MD was not developed and had never been used for the purpose of hybrid execution that we introduce in this work; (iii) MARO brings other empirical advantages over MD, the most evident of those being improved sample efficiency (Figure 2(a)). We elaborate on other advantages in Section 4.4.2.
>
> Additionally, our latest results on SpreadXY4 indicate that our proposed MARO scales better than MD in an environment with more agents. We refer to the second table in the response to reviewer PDRN for the results.
>
> ## More complex dynamic agent behavior
>
> Following the reviewer's suggestion, we start by evaluating the suitability of SMAC to evaluate approaches for hybrid execution in a more complex environment, i.e., as a scenario where the passive sharing of information amongst the agents brings any benefit to task execution.
>
> |           |        Obs.        |        J.Obs       |
> |---------|------------------|------------------|
> |   Return  | 19.7 (-0.02,+0.01) | 19.6 (-0.00,+0.01) |
> | Win rate  | 0.95 (-0.03,+0.02) | 0.93 (-0.01,+0.01) |
>
> Table 1: Average return and win rate (alongside 95\% bootstrapped confidence interval) over five seeds for Obs and J.Obs in the SMAC (2s3z) environment (IQL algorithm). Higher is better.
>
> The results in the above table show that there is no significant performance gap between the fully centralized (Jobs) and fully decentralized (Obs) agents. We already discussed the same phenomenon in the original manuscript, in regards to the result of the agents in the Speaker-Listener environment with QMIX algorithm (Table 1, Page 6). This absence of a performance gap highlights the fact that, in some tasks, there is no benefit to having access to shared information for the decision-making of the agents. In this work we focus on multi-agent cooperative tasks in which the sharing of information is **critical** to their successful execution. We clarified this point in the introduction and evaluation sections of the updated manuscript.
>
> Additionally, we evaluated our approach in scenarios with more agents to explore more complex dynamics: we considered an extension of SpreadXY with 4 agents (SpreadXY4).
>
> |          Obs         |        J. Obs        |          MD         |         MARO        |
> |:--------------------:|:--------------------:|:-------------------:|:-------------------:|
> | -1194.5 (-9.0,+12.0) | -1099.4 (-11.3,+4.1) | -1176.9 (-5.7,+7.1) | -1152.6 (-4.9,+5.0) |
>
> Table 2: Average episodic returns and 95% bootstrapped confidence interval over five seeds for all approaches in the SpreadXY4 environment for IQL. Higher is better.
>
>
> The results of such evaluation show that MARO is able to outperform the MD baseline and exploit passively shared information to perform the task, even in scenarios with higher number of agents.

---

> > ### Author Response · Authors · 2022-11-14
> > **Response to reviewer ymGd (continues)**
> >
> > ## Clarify centralized training and how masks help capturing uncertainty
> >
> > We thank the reviewer for warning us that the centralized training stage is not absolutely clear in the document. Our proposed MARO is composed of two components: a predictive model that replaces observations that other agents fail to communicate by its own predictions and a training scheme that simulates failures in communication. It is possible to use just one of the components of MARO for the problem of hybrid execution, where at test time communication between agents fails with probability $(1-p)$, with unknown $p$ which varies across episodes, as $p$ is sampled from some distribution $\mu$ at the beginning of each episode. The ablated results appear in Table 2. The results show that our centralized training scheme is important for hybrid execution and we clarify how it works. Since we assume no knowledge over the distribution $\mu$ for $p$ at execution time, during the centralized training stage, we sample $p$ at the beginning of each training episode from a uniform distribution. We tested using other distributions and some of the results appear in Figure 3(b). Then, simulating what happens during hybrid execution, we simulate that communication between agents fails with probability $p$, replace the missing observations by the predictions of the predictive model and concatenate a binary vector that indicates whether the agent received communication form other agents. We call that binary vector a mask. For instance, if agent 1 had access to the observations of agent 1 but not of agent 2, the binary vector we concatenate is $(1, 0)$. With the mask, the reinforcement learning controllers have the information to know whether an observation is real (1) or was predicted (0), which may be relevant for choosing the next action to take. Additionally, the masks also allow the recurrent RL controllers to implicitly estimate the degree of observability of the environment, i.e., $p$, and act accordingly. As an example, if the masks received by the RL controller always indicate that the agent is not receiving information from another agent, it can implicitly grasp that the predictions from the predictive model may be accumulating noise, and act accordingly. Empirically, the necessity of the masks can be asserted by looking at the difference in performance between the MD baseline in Table 1 and the ablated version of MARO without predictive model in Table 3. We clarify this question in Section 3.2 of the updated manuscript.
> >
> > ---
> >
> > **We hope that our response has addressed the reviewer's concerns. Please let us know if additional clarifications are required to improve the score.**

---

### Official Review · Reviewer_PDRN · 2022-11-02

**Confidence:** 4
**Correctness:** 3
**Technical Novelty And Significance:** 4
**Empirical Novelty And Significance:** 4
**Recommendation:** 5

**Clarity, Quality, Novelty And Reproducibility:**

Clarity: High. The paper is clear and easy to follow. The prior and related work is properly cited.

Quality: Medium. The paper includes extensive experiments and a detailed analysis of prior work as well as new the new method. However, additional testbeds and analysis would strengthen the work.

Novelty: High. The authors proposed a new paradigm for MARL - hybrid PODPS that can have real-world applications in certain settings.

Reproducibility: High. The code is included alongside the submission.

**Strength And Weaknesses:**

## Strengths

- The paper takes on an interesting problem - hybrid execution in MARL. Although the CTDE regime has gained major attention in recent years, no work has previously studied this paradigm which can potentially have real-world use cases. To me the problem itself is novel.
- The paper includes extensive empirical evaluations and ablation studies that evaluate all aspects of the proposed approach and contrast them with other methods.
- The paper is clear and easy to follow. The contributions of the work can be easily identified.

## Weaknesses

- The major weakness of this work, in my view, is the environments used for empirical evaluation. The authors only have 2D environments with 2-3 agents. It is very unclear if the proposed method would scale to settings with more agents (e.g. 5, 10, 20) or to more complex domains. I think the SMAC benchmark could be an interesting domain for trying out MARO whilst having a large number of agents alongside complex dynamics
- While the work is considerably novel (both the problem and the method), the proposed approach is fairly straightforward. It seems obvious that this approach would outperform a fully decentralized case or MD. So what I would want to see here, which is unfortunately missing, is a thorough empirical evaluation of the method for different communication matrices. For example, empirical analysis of the following settings would strengthen the paper - what happens if
   1. $p{i,j}\neq p_{j,i}$
   2. different $p$ value is used for different co-player pairs
   3. $p$s change throughout the episode
- (Minor) It would be helpful if the authors include screenshots of the environments in the main text.

Note: My score is not final and can change (in either direction) based on the authors' responses or comments of other reviewers.

**Summary Of The Paper:**

This paper proposes a new paradigm for multi-agent reinforcement learning (MARL) whereby the agents are trained in a centralised way but tested in a hybrid fashion. During the hybrid execution, information regarding other agents is hidden during different timesteps, ranging from a fully decentralised mode to a centralized regime. The authors name this problem hybrid-POMDPs and propose a method called MARO to make to most out of the given at times information during execution. The authors provide empirical results that show MARO outperforming other baselines, as well as detailed ablation studies.

**Summary Of The Review:**

This is an interesting work that proposes a new MARL problem that has not been studied before. The authors compare their method with existing approaches and perform an ablation study of its components. However, using more complex environments with more agents would strengthen the work. In addition, I encourage the agents to perform more experiments studying different hybrid execution paradigms, as described above.

---

> ### Author Response · Authors · 2022-11-13
> **Response to reviewer PDRN**
>
> We thank the reviewer for the comments and for acknowledging the novelty of the problem of hybrid execution and the extensiveness of our empirical evaluation.
>
> ---
>
> ## Evaluation in additional scenarios
>
> Following the reviewer's suggestion, we start by evaluating the suitability of a SMAC scenario with 5 agents to evaluate approaches for hybrid execution, i.e., as a scenario where the passive sharing of information amongst the agents brings any benefit to task execution.
>
> |           |        Obs.        |        J.Obs       |
> |---------|------------------|------------------|
> |   Return  | 19.7 (-0.02,+0.01) | 19.6 (-0.00,+0.01) |
> | Win rate  | 0.95 (-0.03,+0.02) | 0.93 (-0.01,+0.01) |
>
> Table 1: Return and win rate (alongside 95% bootstrapped confidence interval) over five seeds for Obs and J.Obs in the SMAC (2s3z) environment (IQL algorithm).
>
>
>
> The results in the table above show that there is no significant performance gap between fully centralized (J.Obs) and fully decentralized (Obs) agents. We already discussed the same phenomenon in the original manuscript, in regards to the results in the Speaker-Listener environment with the QMIX algorithm (Table 1, Page 6). This absence of a performance gap highlights the fact that, in some tasks, there is no benefit to having access to shared information for the decision-making of the agents. In contrast, in this work we focus on multi-agent cooperative tasks in which the sharing of information is \emph{critical} to their successful execution. We clarified this point in the introduction and evaluation sections of the updated manuscript.
>
> |          Obs         |        J. Obs        |          MD         |         MARO        |
> |--------------------|--------------------|-------------------|-------------------|
> | -1194.5 (-9.0,+12.0) | -1099.4 (-11.3,+4.1) | -1176.9 (-5.7,+7.1) | -1152.6 (-4.9,+5.0) |
>
> Table 2: Average episodic returns and 95% bootstrapped confidence interval over five seeds for all approaches in the SpreadXY4 environment for IQL. Higher is better.
>
> Additionally, following the reviewer's suggestion, we also evaluate our approach in scenarios with more agents: we considered an extension of SpreadXY with 4 agents (SpreadXY4). We present such evaluation in the above Table, which show that MARO is able to outperform the MD baseline and exploit passively shared information to perform the task, even in scenarios with higher number of agents.
>
> ## Evaluation with different communication processes
>
> We thank the reviewer for suggesting that we also test other communication processes in hybrid execution. Specifically, the reviewer suggests that we consider asymmetric communication matrices $C$ and dynamic (non-stationary) communication matrices $C_t$. We agree that such experiments would strengthen our work. Therefore, following the reviewer's suggestion, we performed additional experiments where we consider different communication processes. We considered three different settings: a) our default setting $p_{\textrm{default}}$, where $p_{i,j} = p_{j,i} = p$, with $p \sim U(0,1)$ sampled at the beginning of each episode; b) $p_{\textrm{asymmetric}}$, with communication matrices $C$ such that $p_{i, j} \neq p_{j, i}$, where both $p_{i, j}$ and $p_{j, i}$ are independently sampled from a random uniform distribution at the beginning of each episode; c) $p_{\textrm{dynamic}}$, with dynamic communication matrices $C_t$ that are sampled every $5$ time steps. We present the results of this comparison across all environments and RL algorithms in Appendix B.3.3, due to lack of space on the main text. We reproduce here a representative selection of the results, for convenience:
>
> |                           |         SpreadXY + QMIX        |       SpreadBlindf. + IQL      |
> |:-------------------------:|:------------------------------:|:------------------------------:|
> |   $p_{\textrm{default}}$  | -143.0 (-144.9, -141.4) | -405.0 (-407.4, -403.4) |
> | $p_{\textrm{asymmetric}}$ | -141.6 (-143.1, -140.3) | -398.6 (-412.2, -383.3) |
> |   $p_{\textrm{dynamic}}$  | -141.0 (-143.1, -139.0) | -403.6 (-413.6, -395.5) |
>
> Table 3: Average episodic returns and 95% bootstrapped confidence interval over five seeds for different sampling techniques for communication matrices in MARO. Higher is better.
>
>
> The results highlight the robustness of MARO to different sampling choices of the communication matrices between the agents, with no significant performance difference between the three conditions across all environments.
>
> ## Screenshots of the novel environments
>
> We thank the reviewer for the comment. As the novel environments are built on the MPE framework, visually they are quite similar to MPE environments. In Appendix B.1 we have referred to MPE for visual depictions of the environments.
>
> ---
>
> **We hope that our response has addressed the reviewer's concerns. Please let us know if additional clarifications are required to improve the score.**

---

### Author Response · Authors · 2022-11-16
**General response to all reviewers**

We thank all the reviewers for their insightful comments and suggestions. We hope that our responses clarified the concerns that were raised. Please let us know if any further clarifications are required.

---

### Decision · Program_Chairs · 2023-01-20

**Decision:**

Reject

**Justification For Why Not Higher Score:**

There were common concerns around the experimental evaluation, in particular around the simplicity of the benchmarks used and scaling the numbers of agents. Another common concern is that the simple message-dropout baseline performs very similar to the more complicated method suggested by the authors.

**Justification For Why Not Lower Score:**

N/A

**Metareview: Summary, Strengths And Weaknesses:**

Summary:
The authors suggest a new problem setting (hybrid execution), in which each agent at test time can observe the observation of another agent with a fixed probability. The authors also propose a method, MARO, for solving this problem setting using a joint-observation encoder that predicts the observations of other agents.

Strength:
Some reviewers liked the novel setting and appreciated that the paper clearly describes the method.

Weakness:
There were common concerns around the experimental evaluation, in particular around the simplicity of the benchmarks used and scaling the numbers of agents. Another common concern is that the simple message-dropout baseline performs very similar to the more complicated method suggested by the authors.